# Immunotherapeutic Approaches for Glioblastoma Treatment

**DOI:** 10.3390/biomedicines10020427

**Published:** 2022-02-11

**Authors:** Nasser K. Yaghi, Mark R. Gilbert

**Affiliations:** 1Neuro-Oncology Branch, CCR, NCI, National Institutes of Health, Bethesda, MD 20892, USA; yaghink@nih.gov; 2Neurological Surgery, Oregon Health & Science University, Portland, OR 97239, USA

**Keywords:** glioblastoma, GBM, glioma, radiotherapy, surgery, chemotherapy, biologic therapy

## Abstract

Glioblastoma remains a challenging disease to treat, despite well-established standard-of-care treatments, with a median survival consistently of less than 2 years. In this review, we delineate the unique disease-specific challenges for immunotherapies, both brain-related and non-brain-related, which will need to be adequately overcome for the development of effective treatments. We also review current immunotherapy treatments, with a focus on clinical applications, and propose future directions for the field of GBM immunotherapy.

## 1. Introduction

Glioblastoma (GBM), the most common primary adult tumor, remains incurable. Despite well-established standard-of-care therapies consisting of maximal safe tumor resection, concurrent chemotherapy, and radiation followed by additional chemotherapy, the median survival remains consistently less than 2 years [1,2]. Paradoxically, although spread outside the central nervous system (CNS) is rare, even with extensive surgical resection, the infiltrative nature of this disease makes it almost inevitable for recurrence [3,4]. This characteristic of GBM makes immunotherapy a logical consideration, wherein activated immune cells can target even distant infiltrating and isolated tumor cells [5]. In this review prepared for the special issue “Potent Agent Research for Glioblastoma Treatment” in the journal Biomedicines, we highlight biologic aspects of CNS immunology, potential barriers to effective immunotherapy, and promising immunotherapy treatments for GBM. This includes is a detailed focus on glioblastoma and relevant immunotherapeutic treatments, with an emphasis on a brief historical overview in each treatment category, and a curation of more recent treatments, with an evaluation of their efficacy and potential based on the authors’ experience.

## 2. CNS Immune Privilege

Cellular immune response requires dendritic cells, which are the most effective antigen presenting cells (APCs). These cells will phagocytose foreign antigens (including tumor-associated antigens) and then migrate to the cervical-draining lymph nodes (LNs) via lymphatic vessels [6]. The antigen bearing dendritic cells (DCs) will then present these antigens to naïve T lymphocytes (T cells) through major histocompatibility class type I (MHC I) or MHC type II (MHC II) complexes. The challenge with this immune activation process in the CNS has been that, historically, the CNS was thought of as an “immune privileged” organ due to the blood–brain barrier (BBB), the paucity of a lymphatic system, and the paucity of APCs [7,8,9]. However, there is now an accumulation of evidence supporting the function of the “glymphatic system” as an important contributor to the immune response. The glymphatic system is located within the walls of the dural sinuses and ultimately connects to the deep cervical lymph nodes [7,10,11]. Chemotaxis, mediated in part by interferon-gamma (IFNγ) and integrins, allows immune cells to enter the brain parenchyma [12,13]. Additionally, antigens are capable of passing through the walls of cerebral arteries and enter the lymphatics via Virchow–Robin perivascular spaces [14]. In disease states, such as infiltrating glioma, the integrity of the BBB is disrupted, and this may further enhance trafficking of immune cells into and out of the CNS [15]. Therapeutically, the BBB can be disrupted by, for example, hyperosmotic agents, such as mannitol, or focused ultrasound [16,17], to temporarily allow molecules to pass into the brain, and this is exploited as a treatment strategy for super selective intra-arterial infusion of chemotherapeutics [18]. 

## 3. Disease-Specific Challenges for Immunotherapy

### 3.1. Blood–Brain Barrier

The BBB is made up of endothelial cells lining cerebral microvessels, which are linked by tight junctions and have very limited transport of vesicles, only allowing for passive transport of lipid-soluble or low-molecular-weight (<400 Da) molecules [19,20]. In addition, there are astrocytes, pericytes, and the extracellular matrix, as well as active transport systems (e.g., ABC transport system), which contribute to further limiting permeability across this barrier, making drug and particle delivery to the CNS difficult and precluding the use of a large majority of cancer therapeutics for treating most primary brain and spinal cord tumors [21]. Although many primary brain tumors, particularly GBM, have abnormal or leaky vasculature, as a result of rapid angiogenesis, this enhanced permeability and retention (EPR) effect is not as homogenous within the tumor as expected, and many nanoparticles cross the BBB through active transcytosis via endothelial cells [22,23]. 

### 3.2. Cell Trafficking

Immunologically, GBM has been considered to be primarily a “cold” tumor, lacking large numbers of tumor-infiltrating lymphocytes (TILs) [24]. Additionally, the lymphocytes that do make their way into the tumor tend to be exhausted and ineffective. Immune surveillance is critical to the identification and clearance of pathogens. The BBB tightly controls the entry of immune cells into the CNS, and, in a healthy system, there are low numbers of neutrophils and lymphocytes within the parenchyma [25]. The entry of leukocytes into the brain tissue from the blood involves leukocyte adhesion molecules (LAMs), including E-selectin and P-selectin [26]. Interestingly, the expression of these molecules varies by disease, but they are upregulated in neuroinflammatory diseases [27]. With primary malignant brain tumors, parenchymal immune cells vary in both quantity and location. Recent work performed in the Heimberger lab looked at immune-cell populations in various anatomical locations within glioblastomas, using RNA-sequencing data from the Ivy Glioblastoma Atlas Project. They found that T-cell-specific (CD3, CD4, and CD8) and B-cell-specific (CD19 and CD20) marker expression was lower at the leading edge of the tumor, the tumor infiltrating the brain, and the tumor itself, including the necrotic zones, but was notably enriched within the vascular areas of glioblastomas [28]. On the other hand, the myeloid cell population (CD33), microglia, and macrophages (CD68) were uniformly distributed throughout the tumor microenvironment. Additionally, they looked at molecules involved in immune cell attraction in brain tumors and found that cGAS and STING were preferentially expressed in the vascular niches—areas where elevated T-cell numbers could also be detected. These findings suggest that immune-cell trafficking into tumors can be directed and enhanced by chemokines; moreover, a clinical trial using a novel STING agonist in patients with glioblastoma is under development [28].

### 3.3. Microenvironment

The GBM microenvironment is thought to be a major contributor to the paucity of TILs and other components of an anticancer immune response. This immunosuppressive environment results from the production of immunosuppressive cytokines, such as interleukin (IL)-6, IL-10, transforming growth factor-beta (TGFβ), and prostaglandin E_2_ (PGE_2_) [29]. Within the tumor microenvironment, there is inhibition of proliferation of T cells and their effector responses, and there is activation of FoxP3+ regulatory T cells (Tregs). Tumor-associated macrophages (TAMs) and the surrounding tumor microglia release immunosuppressive and pro-tumorigenic cytokines into the tumor microenvironment, inducing cytotoxic T-cell (CTL) apoptosis mediated by PD-L1, CTLA-4, and FasL [9]. This process is further promoted by tumor-associated CD70 and gangliosides that act through both receptor-dependent and independent pathways [30]. The production of immunostimulatory cytokines by the TAMs and microglia is blocked by the interaction of the S100B protein with receptor for advanced glycation end products (RAGE), allowing GBM cells to induce the STAT3 pathway within TAMs [31]. These findings suggest that the S100B–RAGE interaction may be vital in STAT3 activation and ultimately result in macrophage and microglia suppression in gliomas. 

As described above, the GBM microenvironment is a complex interaction of tumor and host cells with equally complicated cellular and humoral interactions. Several aspects of the signaling pathways and chemical modulators warrant further description. The STAT3 signal transduction pathway is critical in the interaction of tumor cells with the microenvironment. STAT3 is activated in a process involving the IL-6 family of cytokine receptors [32]. STAT3 has been implicated through multiple lines of evidence to play a pro-tumorigenic role in the GBM microenvironment. STAT3 is upregulated in GBM in hypoxic conditions, resulting in increased expression of vascular endothelial growth factor (VEGF) and hypoxic inducible factor-1 (HIF-1), and STAT3 has been noted to be critical in maintaining tumor stem cells [33,34]. STAT3 has also been shown to be capable of inactivating innate and adaptive immune responses, along with the induction of immune tolerance via Tregs [35,36,37]. The hypoxic microenvironment around tumors induces CNS macrophages to become TAMs, which subsequently adopt a tumor-supportive phenotype (M2 macrophages), and this process is mediated by the STAT3 pathway [38].

Although there are many cytokines and chemokines secreted in the GBM microenvironment, IL-10 is thought to be a key immunosuppressive cytokine and is found in high quantities in various neoplasms and primarily secreted by macrophages, as well as GBM cells [39,40]. IL-10 in the tumor microenvironment inhibits the production of IFNγ and TNFα; it also downregulates MHC class II in monocytes, establishing TIL anergy, and, in doing so, it enhances tumor growth [41,42]. PD-L1 expression on microglia cells is upregulated when the cells are near GBM cells, resulting in T-cell apoptosis [43]. These findings underscore the largely inhospitable immune microenvironment, leading to the consideration of targeting aspects of the tumor microenvironment to reverse tumor-mediated immune suppression and enhance the efficacy of the innate immune response and other immunotherapies.

### 3.4. Cancer Stem-Cell Niche

GBM stem cells (GSCs), also referred to as glioma-initiating cells (GICs), are a heterogeneous population of cells which are multipotent, undifferentiated, have the capacity for self-renewal, and are highly tumorigenic [44,45]. GSCs mediate the adaptive immune system T-cell responses via the STAT3 pathway, which, as previously discussed, is a key regulator of tumor-mediated immune suppression [46]. The GSCs produce macrophage inhibitory cytokine-1 (MIC-1), which is a predictor of poorer outcome [47]. Additionally, monocytes can be converted into the immunosuppressive macrophage (M2) phenotype by GSCs, which also recruit monocytes into the tumor microenvironment [48]. While there is not a complete consensus, GSCs can be identified by various surface markers or molecular mediators, including CD133, CD90, CD44, L1CAM, A2B5, and GPD1 [49]. This lack of consensus on marker definition for GSCs creates some challenges. However, they can be characterized by their function, including in vivo tumorigenic potential and pluripotent nature [50]. New and more accurate methods for the effective isolation and identification of GSCs are needed. For example, CD133, also known as PROM-1, has been the longest used and most validated marker for GSCs [51]. However, it has been shown that some CD133-negative cells also have the ability to initiate tumors; therefore, CD133 alone may not be a reliable marker for GSC [52].

The GSC niche is a highly heterogeneous and complex environment, with contributions from macrophages, pericytes, astrocytes, endothelial cells, and fibroblasts. In this context, studies suggest that there are three major GSC niches, namely the perivascular niche, perinecrotic/hypoxic niche, and immune niche [53]. The perivascular niche is characterized by poor and irregularly formed leaky and friable blood vessels, and GSCs act to regulate this vasculature and promote tumor angiogenesis [54,55]. 

Likewise, hypoxia, a hallmark of GBM, is most notable within necrotic tumor regions where oxygen levels are very low [56]. One of the regulators of this hypoxic tumor environment, HIF-2α, plays an important role in maintaining the GCSs, and when this gene is silenced, GCS function is compromised [57]. Interestingly HIF-1α is induced by extreme hypoxia levels, ~1% O_2_, while HIF-2α is induced slower in response to moderate hypoxia levels. Importantly, HIF-1α is found in both CSCs and non-stem tumor cells, while HIF-2α is exclusively found in CSC and not detected in the normal human macrophages, thus making it a potential great target for treatments [58].

The immune niche, notable for the preponderance of TAMs, the most prevalent inflammatory cells within GBM, plays an important role in the maintenance of GCSs by the pleiotrophin–PTPRZ1 signaling axis [59,60]. GSCs exhibit a relative radioresistance compared with other, more differentiated tumor cells, and this is likely the result of more efficient DNA double-strand break repair mechanisms; GSCs have greater resistance to chemotherapeutic agents, in part, due to overexpression of ABC transporter proteins [50,61,62]. 

### 3.5. Glioma–Neuronal Interactions

Although there is increasing recognition that neuronal–glioma interactions are an important aspect of tumor biology and cancer growth, the impacts of these interactions on tumor progression and microenvironment are just now being investigated. The Monje Lab at Stanford has performed significant work on the electrical and synaptic integration of glioma in neural circuits. Synaptic connections exist between neurons and normal oligodendroglia precursors cells (OPCs), and signaling at these synapses can influence the proliferation and survival of the OPCs [63]. Analysis using single-cell transcriptomic datasets from pretreatment biopsy samples of adult and pediatric high-grade gliomas revealed synaptic gene enrichment within subpopulations of malignant cells, as is consistent with the theory that malignant cell populations assume distinct roles in the cancer ecosystem [64]. Additionally, using electron microscopy to visualize the ultrastructure of the synapse and electrophysiologic measurements, glioma excitatory postsynaptic currents provided functional evidence supporting the neuron-to-glioma synapses which could be blocked by tetrodotoxin (TTX), a voltage-gated sodium channel blocker [64]. The impact of depolarization on cell behavior was examined by using very sophisticated optogenetic techniques. Most notably, it has been shown in a xenographed glioma cell model that glioma depolarization robustly promoted xenograft glioma proliferation when compared with mock stimulated controls [64,65,66]. Additionally, in bidirectional neuron–glioma interactions, gliomas are thought to increase neuronal activity, while neuronal activity increases glioma growth [67]. This is specifically important, given the high frequency of seizures associated with intraxial tumors, and it has been confirmed that glioma induces neuronal hyperexcitability and seizures [68]. The Monje group also assessed hyperexcitability in primary human glioblastoma intraoperatively, using electrocorticography in awake patients with cortical high-grade gliomas prior to resection [64]. When measuring outside the nodular necrotic core of the tumor, they found a significant increase in high gamma band power in tumor-infiltrated brain compared to normal brain tissue [64]. In summary, these findings underscore the increasingly recognized complexity of the glioma microenvironment with the impact of neuronal activity as a regulator of glioma progression. 

Other studies are using magnetoencephalography imaginary coherence measures of functional connectivity to identify intratumoral high (HFC) and low (LFC) functional connectivity network hubs in patients with dominant temporal lobe GBM [69]. They found that TSP1 mediates glioma-induced neuronal synaptogenesis and supports tumor functional network integration that, overall, negatively impacts behavior and survival. More work in this area is needed to understand the full implications of glioma–neuronal functional interactions. 

### 3.6. Iatrogenic Impact

Dexamethasone is a corticosteroid and commonly used in the management paradigm for brain tumors treating vasogenic cerebral edema frequently associated with intracranial lesions [70]. Corticosteroids are the primary therapy for immune-related adverse events (irAE) that develop with cancer immunotherapy, particularly immune checkpoint inhibitors. This immunosuppressive effect was established in 1976, a finding particularly pertaining to intracranical tumors that has significant implications for the effectiveness of immunotherapy in the treatment of GBM [71]. Fauci et al. measured absolute circulating lymphocytes and monocytes up to 48 h after standard treatment dose of either dexamethasone, hydrocortisone, or prednisone and found a marked though transient decrease in lymphocytes and monocytes maximal at 4–6 h post-administration, with subsequent return to normal counts or rebound to slightly supranormal counts by 24 h [71]. 

A group in Western Australia calibrated dexamethasone dosing in mice to the equivalent lymphocyte depletion seen in patients with cancer and found that, while peripheral blood T and B lymphocytes, along with NK cells, were depleted, the same cells were unchanged within tumors [72]. Additionally, they showed that the immune checkpoint molecules PD-1, OX40, GITR, and TIM3 on TILs were unaltered with dexamethasone treatment [72]. Despite these findings, our group has shown that dexamethasone diminishes naïve T cells’ ability to proliferate and differentiate by attenuating the CD28 co-stimulatory pathway [73]. This inhibitory effect can be reversed with CTLA-4 blockade restoring, in part, the T-cell ability to proliferation in the presence of dexamethasone, and this enhanced survival in tumor-bearing mice. There is also evidence supporting differential effects on B and T lymphocytes, as well as NK cells, in humans with hydrocortisone treatment. Moreover, there is evidence of significant changes in circulating human cytokine levels, including leptin, C-peptide, GIP, and insulin being increased, and the majority, including PAI-1, MCP-1, IP-10, MIG, CTACK, MIP-1a, MIP-1b, TRAIL, eotaxin, IL-1b, IL-8, and FGF-basic, being decreased [74]. 

The effects of systemically administered hydrocortisone have been shown to change the gene expression in human peripheral blood mononuclear cells (PMBCs) [74]. Gene-expression microarray studies have shown that there were qualitatively different effects with different doses of hydrocortisone. Interestingly, gene-expression changes were observed as early as one hour after hydrocortisone infusion, with the peak frequency of gene-expression changes after four hours and with more gene expression changes at higher hydrocortisone dose compared with lower dose [74]. After four hours of hydrocortisone infusion, there were progressively fewer gene-expression changes [74]. Overall, hydrocortisone downregulates gene sets (modules derived from the Modular Analysis Framework) associated with inflammation and cell death, NF-κB, and Toll-like receptor (TLR), while apoptotic signaling transcripts and cell-cycle-related mRNAs were upregulated [74]. The duration of corticosteroid effect on immune cells is not known, but investigations are underway to address this question, as this will impact the use of immunotherapy in most patients with brain tumors who often require corticosteroids during their illness.

Immunotherapy studies for GBM are often combined with chemotherapy, most commonly temozolomide. The concurrent use of temozolomide with immunotherapy does raise concerns, as temozolomide often causes profound lymphopenia. Paradoxically, some studies have demonstrated that this lymphopenia can allow for the expansion of CAR T cells or immunotherapeutic vaccines [75]. Preclinical studies in a murine model of GBM showed that dose-intensified TMZ (higher than standard of care TMZ dosing) promoted a dramatic CAR T-cell proliferation and enhanced persistence in circulation compared to CAR T cells alone or standard-of-care TMZ dosing [76]. Karachi et al. further examined TMZ induced lymphopenia and found standard TMZ dosing reduced both CD4+ and CD8+ T cells, and results in greater CD8+ T-cell exhaustion and overall poorer outcomes with PD-1 antibody treatment compared to lower/metronomic TMZ dosing, which maintained cytotoxic T-cell activity and direct tumor killing [77]. Additionally, preclinical studies have observed reduced PD-L1 expression on GBM cells following TMZ treatment, suggesting that chemotherapeutic intervention reduces the immunosuppressive profile of GBM [78]. 

Compared to more classical chemotherapies, new cancer drugs, namely immunotherapies, are generally less toxic and possibly more tolerable for the patient [79]. The most common side effects include non-specific constitutional symptoms, including gastrointestinal (GI) discomfort, mucositis, and myelosuppression, which are mainly mild or absent. However, there are life-threatening side effects, which include inhibition of angiogenetic pathways, severe inflammatory syndromes, and autoimmune disorders [79]. Cytokine-release syndrome (CRS) is a potentially life-threatening systemic inflammatory reaction that is characterized by fever, chills, hypotension, and tachycardia during or immediately after drug administration [79]. This reaction appears to be mediated by immensely high levels of IL-6, which is released after the activation and cytotoxic damage of different lymphocyte populations [80]. In addition, though the mechanism is currently unclear, cytokine release has been implicated in neurologic events post-treatment, including tremor, encephalopathy, cerebellar alteration, or seizures, and lethal cerebral edema has also been observed, as well [81]. Immune related adverse events (IRAEs) have been reported in up to 85% of patients treated with ipilimumab and up to 70% of patients treated with PD1 axis inhibitors [82,83]. In addition to the most common symptoms, diarrhea and enterocolitis, potentially requiring ICU admission, nonspecific symptoms, such as weakness, fatigue, confusion, and nausea, can lead to the diagnosis of complex hormonal disorders involving the thyroid and pituitary glands [84,85]. There are data from clinical trials, specifically with ICIs, which report increased incidence of hypophysitis with CTLA4 inhibition and thyroid dysfunction with PD-(L)1 inhibition [86]. Interestingly, there have also been a few cases of type 1 diabetes mellitus and primary adrenal insufficiency noted [86]. Further research to understand, in more detail, the mechanisms of IRAEs will be critical to the continued use of immunotherapies to ensure patient safety. 

### 3.7. Interpretation of Tumor Imaging

The successful implementation of immunotherapy for brain tumors, particularly for GBM, may paradoxically result in a challenge in distinguishing treatment response from tumor progression. For example, in serial sampling of cancers, such as melanoma, where the use of immune checkpoint inhibitors (ICIs) has definitive efficacy, tumors show extensive infiltration of immune cells and other histologic hallmarks of an inflammatory response. Whereas the robust lymphatic system in most organs facilitate tumor clearance, these structures are rudimentary in the CNS; hence, a treatment-induced immune response may evoke edema, increased blood perfusion, increased contrast enhancement, and mass effect. All of these changes are characteristic of tumor progression. However, there have only been limited series demonstrating this challenge of immune pseudoprogression with ICI use in glioblastoma [87]. Some studies suggest that patients on immune checkpoint inhibitors (ICIs) or other targeted immunotherapies have been shown to develop radiation necrosis following SRS at a much higher rate than patients on chemotherapy or no systemic therapy [88,89]. Additionally, ongoing investigations are attempting to characterize tumor immune states by radiomic features, including sphericity, heterogeneity, and sharpness of borders with predicted response to ICIs [90]; and composite models using AI and deep learning to combine radiomic models with other variables, including clinical and molecular features, are promising. Immune-cell trafficking, as described above, is a critical issue for brain-tumor immunotherapy. Recent studies suggest that imaging technologies can be used to non-invasively monitor the kinetic infiltration and therapeutic efficacy of chimeric antigen receptor (CAR) T cells in GBM with MRI, specifically using ultra-small superparamagnetic particles of iron oxide (USPIOs), to label CAR T cells [91]. These nanoparticles did not appear to exert any negative effects on the CAR T cells’ efficiency and allowed for early validation of CAR T-cell therapeutic effect in solid tumors. A summary of disease-specific challenges for immunotherapy is presented in Figure 1. 

## 4. Immunotherapy

### 4.1. Checkpoint Inhibition 

ICIs in preclinical GBM models have shown promising therapeutic activity; however, clinical trials have yet to show significant survival benefit in newly diagnosed or recurrent GBM [92]. ICIs are a class of immunotherapy that acts to remove the inhibitory brakes of the T cells, thereby activating the immune system and allowing for a more robust antitumor responses [93]. Currently, ICIs in clinical use are monoclonal antibodies directed to target surface receptors, although small molecule inhibitors are in development. Blocking the surface receptors with antibody or inhibiting the signaling pathway directly with a small molecule is designed to reverse the inactivation or exhaustion of T cells. With these pathways blocked or attenuated, the immune system can generate a stronger antitumor response. The United States Food and Drug Administration (FDA) has approved programmed cell death 1 (PD-1) or ligands PD-L1/PD-L2, along with cytotoxic T-lymphocyte-associated protein 4 (CTLA-4), for use in humans. There are other ICIs in development targeting different immune checkpoints, including T-cell immunoreceptor with immunoglobulin and ITIM domain (TIGIT), T-cell immunoglobulin- and mucin domain-3-containing molecule 3 (TIM3), CD73, adenosine A2a receptor, and lymphocyte-activation gene 3 (LAG3). 

The two most widely used ICIs are anti-CTLA-4 and anti-PD-1/anti-PD-L1 [93]. These rely on the concept of reversal of immunosuppression. Anti-CTLA-4 (i.e., ipilimumab) blocks the ability of CTLA-4 to bind with greater affinity to B7, thereby allowing B7 to continue to participate in co-stimulation of T cells via CD28, preventing T-cell downregulation and deactivation [92]. PD-1 is expressed on a host of activated immune cells, and PD-L1 and PD-L2 are expressed on APCs and cancer cells. When engaged, this suppresses T-cell activity, therefore blocking this interaction and allowing T cells to remain active [92]. Anti-PD1 ICIs include cemiplimab, nivolumab, and pembrolizumab, while anti-PD-L1 ICIs include duralumab, atezolizumab, and avelumab. 

The expression levels of PD-1 in tumors may help predict response to anti-PD-1 and anti-PD-L1 therapy, although, despite elevated expression, some cancers do not respond [94]. Additionally important, in the absence of ICI treatment, PD-L1 expression appears to correlate with overall prognosis, with higher expression of PD-L1 being correlated with worse overall outcome [95]. The tumor mutational burden has also been correlated with response to ICIs; however, this relationship is not absolute, as there are tumors (i.e., Merkel Cell Carcinoma) that have a low mutational burden, yet a very high response to ICIs. GBM typically has a modest mutational burden, although IDH mutated gliomas may develop a hypermutated phenotype after extensive exposure to alkylating agent chemotherapy [96,97]. There is a phase II clinical trial that is specifically designed to look at the effectiveness of nivolumab in patients with IDH1 or IDH2 hypermutated gliomas (hypermutator phenotype) currently recruiting (NCT03718767). 

Despite improvements in survival of patients with other cancers, ICIs have yet to show the same benefit in the treatment of GBM. CheckMate-143 (NCT02017717), a phase III trial comparing the PD-1 inhibitor nivolumab to VEGF-A inhibitor bevacizumab, also showed no improvement in survival in patients with recurrent GBM [98]. Another clinical trial, Keynote-028 trial (NCT02054806), investigated the efficacy of pembrolizumab, an anti-PD1 ICI, in different advanced solid tumors, including GBM, which showed modest benefit [99]. There is currently a host of combined treatments with ICIs in clinical trial evaluation; such treatments are discussed later in Section 4.5. The initial preclinical success of single-agent therapy with immune checkpoint inhibitors has not yet translated into the same efficacy in larger randomized trials. For this reason, we believe that identifying upfront at the outset of a clinical trial which patients will benefit from ICI will allow this promising therapy to be used most optimally. A phase II clinical trial is currently accruing for patients with newly diagnosed glioblastoma or gliosarcoma treated who will receive ICIs after chemoradiation, with the aim to identify an association of peripheral blood immunologic response to therapeutic response to ICIs (NCT04817254).

### 4.2. Vaccine Therapy 

Glioblastoma is not considered to be immunogenic; it does not generate a spontaneous innate immune response, as evidenced by the typical paucity of immune cells in tumor specimens. Vaccine therapy attempts to alter this by providing tumor-cell-specific immune stimulation, thereby enhancing the adaptive immune system to target the cancer. Dendritic cells have been used for vaccines (DCVs). DCs are the professional antigen-presenting cells that reside in tissues [100]. When pathological changes in the tissue are detected, DCs will migrate to the draining lymph nodes and will present processed antigenic material via human leukocyte antigen (HLA) class I and II molecules [101]. This process then allows antigen-specific cytotoxic T lymphocytes (CTLs) and helper T cells (Th) to get activated, proliferate, and, subsequently, perform effector functions. DCVs seek to exploit this process by vaccinating patients with DCs loaded with tumor-associated antigens (TAAs), with the aim that they migrate to local lymph nodes and initiate targeted antitumoral T-cell response [102]. If effective, these T-cells would selectively kill tumor cells and help generate immunologic memory to aid in the prevention of recurrence [103]. 

Dendritic cells vaccines (DCVs) were first applied in the treatment for B-cell lymphoma in 1996 [104]. In 2000, DCVs showed promise in animal models of glioblastoma [105]. For example, mice vaccinated with DC cells that had been pulsed with lysates from a glioma cell line transfected with EGFRvIII produced an immunologic memory response that protected mice from subsequent intracranial tumor challenge and significantly prolonged survival [105]. The first clinical efficacy of DCVs was seen in 2006 in patients with hormone refractory prostate cancer, using DCs loaded with a fusion protein of granulocyte-macrophage colony-stimulating factor (GM-CSF) and prostatic acid phosphatase (Provenge/sipuleucel-T) [106]. The median overall survival in vaccinated patients was significantly improved with DCV.

Liau and colleagues have pioneered the use of DCV as active immunotherapy for GBM [107]. Several phase I and II studies have been reported and, as summarized in a recent review, a randomized double-blinded phase III trial has been completed, although the results have not yet been published [102]. Most of the patients in these trials had newly diagnosed or recurrent GBM; however, some studies did include grade III tumors. It is not yet clear if the grade of tumor effects the immunological response rate, although newly diagnosed patients generated a more robust immune response measured in peripheral blood [108]. These studies also indicate that the vaccine is well tolerated, with rare serious adverse events and controllable symptoms, including lymph-node swelling, pain, induration, erythema, and itching at injection site, along with myalgias, headache, fatigue, and fever [109]. There is growing interest in combining vaccine strategies, including DCVs with other immunotherapies, such as immune checkpoint inhibitors, with the premise that the vaccine may help focus the ICI-related augmentation of the immune system [102]. 

The efficacy of DCVs is highly dependent on the antigenic targets chosen, specifically tumor-associated antigens (TAAs) or neoantigens of the individual tumor. In previous DCV studies, antigens such as apoptotic bodies of tumor cells, tumor lysates, irradiated tumor cells, tumor mRNA, and peptides from tumor cell surface have been used [102]. While EGFRvIII remains the most prominent TAA tested to date, there are other antigens in development [110]. For example, isocitrate dehydrogenase (IDH), which is not normally found in human cells and almost always found in tumor cells, is a promising TAA [111]. Specifically, the IDH1 mutation R132H is found in secondary low-grade gliomas, and peptide vaccines targeting this mutation are currently in phase I clinical trials [112]. Although the mutational burden of GBM is relatively low, there is significant heterogeneity between glioblastomas, but strategies utilizing individualized vaccines may overcome issues related to heterogeneity, thereby providing a potential route to personalized medicine treatments. There are multiple studies that find novel TAAs unique to a patient’s tumor by comparing the whole exon sequence data from the resected tumor and matched normal tissue. A group of these antigens that are predicted to have strong binding affinity for HLA type-I are then used to develop a personalized vaccine [113].

Recent studies suggest that TAA from whole-tumor cells may be most optimal for presentation/cross-presentation via HLA class I and II molecules to help reduce the risk of immune escape from loss of TAA variants [114]. However, TAAs only represent a small fraction of total tumor-cell proteins, and low tumor content produces an even smaller quantity of TAAs. Therefore, more extensive tumor resection, potentially enhanced with the use of fluorescence-guided surgery (FGS), as well as appropriate tumor tissue handling, should be able to provide adequate samples for TAA extraction [115]. As an alternative to whole-tumor sources of TAAs, molecularly defined TAA can be used and DCs can be transfected with the specific mRNA for the desired target antigen [116]. It is not currently known if molecularly defined TAA or whole-tumor cell sources are most optimal for generating antitumoral immune response in GBM [117]. However, TAAs alone do not influence efficacy, as other factors, such as the ability of DCs to differentiate and mature, are also dependent on the immunosuppressive factors produced by the tumors cells and are a determinant of immunoreactivity [118]. 

### 4.3. Oncolytic Viral Therapies 

Oncolytic viruses (OVs) are viruses designed to selectively infect cancer cells, and, subsequently, by hijacking intracellular processes, lead to cell lysis and the release of active virus, as well as tumor antigens. This key characteristic promotes the propagation of the tumor-cell killing, as well as augmenting the host immune response [119]. While the early OVs functioned solely by lysing tumor cells, more recent OVs have been designed to act as vectors of delivery for genetic material payloads intratumorally, including genes for neoantigens or key cytokines for microenvironment regulation [120]. Different viruses, including HSV, parvovirus, adenovirus, measles, and replicating retroviral vectors, have been tested both preclinically and in early phase clinical trials. There are recent phase I/II trials for GBM which have shown promising results, with a small subset of patients achieving long-term survival of over 3 years; these include measles virus MV-CEA [121], adenovirus DNX-2401 (Ad5-delta24-RGD) [122], polio-rhinovirus chimera (PVSRIPO) [123], parvovirus H-1 (ParvOryx) [124], and retroviral vector Toca 511 (vocimagene amiretrorepvec and Toca FC) [125]. There is, to our knowledge, only one phase III randomized controlled trial for the retroviral vector Toca 511, which is a murine leukemia virus (vocimagene amiretrorepvec) with flucytosine (Toca FC) comparing to the standard of care (NCT02414165) [126]. This combined phase II/III clinical trial involved four centers and randomized 403 patients with first- or second-recurrence GBM or anaplastic astrocytoma to treatment with delivery directly to resection cavity. The primary outcomes measured were overall survival and secondary outcomes, which included safety, dural response rate, duration of durable response rate, durable clinical benefit rate, and overall survival. The authors found no overall survival or other efficacy benefit with Toca 511 and Toca FC treatment [126]. 

In May 2016, the FDA granted breakthrough therapy designation for the recombinant oncolytic poliovirus, polio-rhinovirus chimera (PVSRIPO). A phase I dose escalation study was performed (NCT01491893) with intratumoral delivery of virus in patients with recurrent supratentorial grade IV malignant glioma [123]. The safely profile was concerning, with 19% of participates sustaining grade 3 or higher adverse events; however, 20% of patients had long term survival for 57–70 months after the PVSRIPO infusion [123]. PVSRIPO has subsequently moved on to a phase II randomized study (NCT02986178). Despite the report of a subset of long-term survivors in these early phase clinical trials testing OVs, a meta-analysis of OVs trials for recurrent GBM showed that 2- and 3-year survival rates were not statistically different from the standard of care [127]. A final determination of the benefit of oncolytic viral therapy awaits larger randomized phase II/III trials. 

### 4.4. Chimeric Antigen Receptor (CAR) T Cells 

The use of chimeric antigen receptor (CAR) T cells has its origins in the use of adoptive cell transfer (ACT), using autologous lymphocytes, in the treatment of metastatic melanoma [128]. The use of ACT-based therapy for GBM began almost 40 years ago with the intratumorally injection of autologous lymphocytes; however, this approach did not provide specificity to the brain-tumor target [129,130]. T-cell receptors (TCRs) were also being genetically engineered and found to generate robust T-cell responses, leading to the emergence of CAR T-cell therapy [131]. In 2017, a CAR T cell directed against CD19 (tisagenlecleucel) was FDA approved for the treatment of patients with relapsed or refractory B cell acute lymphoblastic leukemia; treatment generated a 63% complete remission rate [132]. 

To further develop this treatment, mechanisms making T cells more specific in their targeting ability are being developed, and genetically engineered T cells that were designed to target TAAs have shown efficacy in preclinical studies. For example, the first CAR T-cell targeted therapy was directed at interleukin-13 receptor alpha 2 (IL13Ra2), which is overexpressed in GBM and not found in normal tissue [133]. Furthermore, this receptor is known to be expressed on GSCs, and a decrease in glioma-initiating activity was observed after CAR T-cell treatment in an orthotopic mouse tumor model. Since then, many more targets have been used in the design of CAR T cells, including human epidermal growth factor 2 (HER2), erythropoietin-producing hepatocellular carcinoma A2 (EphA2), and epidermal growth factor receptor variant III (EGFRvIII), which is expressed in a subset of GBM and not in healthy brains [134,135,136]. 

First-generation CAR T cells were a synthetic molecule with an antigen-recognition domain (ectodomain), and this was linked to an endodomain consisting of a CD3 activation domain (first generation), and subsequently one or two co-stimulatory domains, such as CD28, 4-1BB, or OX40 (second and third generation) [137]. Additionally, CAR T cells can bind to TAAs unrestricted to MHC class I expression, especially important given tumor cells frequently loose MHC class I expression allowing escape from T-cell immunity [138]. 

While CAR T cells do show promise in the treatment of brain tumors, the creation of CARs with specific novel tumor targets is very time-consuming and challenging. Therefore, there is some interest in creating universal CARs (uCARs). These CAR vectors are engineered to express an antigen-recognition domain specific for fluorescein isothiocyanagte (FITC), and these CAR T cells can bind to FITC-tagged monoclonal antibodies to HER2 or CD20 [139]. The efficacy of these uCARs has not been fully evaluated. Additionally, given the significant tumor heterogeneity and plasticity of GBM, there has also been interest in increasing the number of targets for each CAR T cell to prevent antigen escape. There are a host of multitarget CARs, including bi-specific (co-expressed or pooled), trivalent, tandem, and split CARs [137]. For example, there has been a design of HER2 and IL13Ra2 co-expressing CAR T cell that has shown improved antitumor responses compared with the individually expressing CAR T cells [140]. 

As CAR T-cell therapy entered clinical trials, there have been safety and toxicity concerns, particularly cytokine release syndrome (CRS). This toxicity is characterized by the rapid and sustained cytokine release of particularly inflammatory cytokines, IL-6, IL-2, IL-10, and IFNγ after activation of the T cells [141]. Of note, an anti-IL-6 antibody, tocilizumab, has been developed and is FDA approved for the rapid reversal of severe CRS syndromes [142]. 

Almost all the CAR T-cell targets are being evaluated in early phase clinical trials, including, HER2, EGFRvIII, IL13Ra2, EphA2, GD2, B7-H3, and chlorotoxin [137]. Additionally, as the field moves toward understanding the value and clinical benefit of combinational therapies, there is currently a clinical trial evaluating IL13Ra-2 CAR T cells combined with ICIs nivolumab and ipilimumab (NCT04003649). Many of these early phase studies have shown limited-to-no clinical benefit, and some shown severe toxicities. There are other potential unique targets for CAR T-cell therapy directed at GBM, including CD70, which is often overexpressed in the isocitrate dehydrogenase (IDH) wild-type gliomas and is associated with poor survival [143]. CD133, a marker of CSCs and self-renewal, also associated with high radiotherapy and chemotherapy resistance, is another CAR T-cell target [144]. Lastly, MET, a receptor of hepatocyte growth factor (HGF) that is overexpressed in GBM, may also be a good CAR T-cell target [145]. With multiple clinical trials currently in the recruitment phase, it is too early to determine the overall clinical efficacy of CAR T-cell therapy. 

### 4.5. Combinatorial Approaches

While there have been many advances in the field of immunotherapy for GBM, there has been an overall lack of breakthrough efficacy in clinical trials. This has garnered interest in examining combinatorial approaches to help not only target the immune system to the tumor but also to reverse tumor-mediated immune suppression or prevent antigen escape. These combined approaches will also involve radiation, as preclinical data have demonstrated the synergistic effects of stereotactic radiotherapy, which, in part, functions to release tumor antigens [146,147]. 

However, combined strategies involving ICIs for GBM have remained largely without clear therapeutic benefit [148]. The discordance between preclinical- and clinical-trial studies is partially the result of murine models not effectively recapitulating the heterogeneity of human GBM. Additionally, the relatively long duration of human cancer compared with the duration of murine models (typically measured in a few weeks) may prevent the mouse systems from emulating phenomena such “exhausted T cells”, particularly TILs, which have upregulated immune checkpoint molecules, including CTLA-4, PD-1, Indoleamine 2,3-dioxygenase (IDO1), T-cell immunoglobulin and mucin-domain containing-3 (TIM-3), and Lymphocyte-activation gene 3 (LAG-3) [149]. Therefore, for human cancers, combinatorial strategies may be required, as a single immune checkpoint inhibitor will likely be insufficient to reverse immune suppression if other evasion pathways remain functional. Combination strategies for other cancers, such as metastatic melanoma, have been successful, with approval by the FDA in 2016 [149]. Preclinical findings support the use a triple checkpoint blockade, including Indoleamine 2,3-dioxygenase 1 (IDO1) as a target. IDO1 is a checkpoint molecule that is implicated in immunosuppression in GBM and found to be associated with Treg infiltration and poorer clinical outcomes [150]. Combining anti-IDO1 antibodies with anti-CTLA-4 and anti-PD-L1 antibodies decreases the presence of tumor-infiltrating Tregs and provides a durable survival benefit [151]. There is a phase I clinical trial currently recruiting to test the combination of PD-1 and IDO1 dual blockade (NCT04047706). Likewise, there is also a phase I clinical trial involving the combination of anti-PD-1 and anti-LAG-3 ICIs that is currently active but not recruiting (NCT02658981). Combinations of ICIs with costimulatory molecules are another paradigm being investigated, specifically regarding CD137 (4-1BB), which is a costimulatory molecular that is implicated in the activation and infiltration of cytotoxic T lymphocytes into tumor sites [152]. Preclinical studies combining anti-CTLA-4, radiation and anti-CD137 have shown very promising results [153], and, in recurrent GBM patients, the ABTC 1501 trial involving anti-LAG-3 or anti-CD137 alone and in combination with anti-PD-1 has preliminarily shown improved overall survival in the combined anti-PD-1 and anti-CD137 group [154].

ICIs are also combined with radiation radiotherapy, given some evidence that radiation may enhance immunogenicity [148]. This enhanced immune response is characterized, in part, by an increased number of antigen-experienced T cells and effector memory T cells, with greater infiltration ability of these T cells into tumors, along with increased immunogenicity of APCs with upregulated tumor-associated antigen-major histocompatibility complexes [155]. Combined ICIs with radiation have been examined in patients with metastatic melanoma, in which the mechanism of action of both treatments was immunologically non-redundant, with radiation diversity in the TCR repertoire and ICI promoting T-cell-effector responses [156]. 

ICIs can be combined with vaccines to counteract immune resistance contributed to, in part, by poor antigen priming. Improved antigen priming synergistically enhances both antigen recognition and T-cell effector function [157]. Combining adjuvant vaccination and an anti-PD-1 antibody has displayed success in stage IIIC and IV high-risk melanoma, enabling relapse-free survival [158]. Additionally preclinical and clinical data support DC vaccination results in the upregulation of PD-1 expression on T cells, and DC vaccination combined with anti-PD-1 increases the expression of memory markers, in addition to integrin homing markers seen in TILs [159]. To by bypass the blood–brain barrier, local delivery of chemotherapeutic agents, including carmustine (BCNU) or TMZ, through biodegradable wafers or thermo-responsive biodegradable pastes has been used; and now there is increased interest in the delivery of ICIs in the same fashion [160,161]. Similarly, there is an ongoing examination in a phase I clinical trial of the combination of intratumoral ipilimumab at the time of surgical resection with systemic nivolumab for recurrent GBM (NCT03233152). 

For patients with newly diagnosed GBM, two phase III trials have recently been completed. The CheckMate-548 trial examined temozolomide plus radiotherapy combined with nivolumab or placebo in patients with newly diagnosed MGMT-methylated GBM (NCT02667587), and the CheckMate-498 trial is investigating nivolumab versus temozolomide, in combination with radiotherapy, in patients with newly diagnosed MGMT unmethylated GBM (NCT02617589). Unfortunately, neither study demonstrated a survival improvement with the addition of nivolumab. Concerns regarding patient selection, particularly corticosteroid use abrogating ICI efficacy, led to the previously mentioned clinical trial NCT04817254, in which patients’ response in peripheral blood is being monitored. The study will attempt to determine if this peripheral blood response is associated with an improvement in survival. This finding would support the efficacy of ICIs in GBM, recognizing that not all patients are able to mount an immune response. A summary of GBM immunotherapy approaches is presented in Figure 2. 

## 5. Biomarkers in Immunotherapy

Given the lack of success of immunotherapy in GBM, there are also limited predictive biomarkers that are in use for patient selection and monitoring of therapy. These are summarized in Table 1. Currently, the Immunotherapy Response Assessment for Neuro-Oncology (iRANO) uses MRI to differentiate responders from non-responders [162]; however, imaging alone cannot accurately predict response to treatment [87]. There are several putative predictive markers from immunotherapy studies in other cancers, including tumor expression of PD-L1, tumor mutational burden, and loss of microsatellite stability [163]. For example, in gastric cancer, tumor-infiltrating CD8+ T cells and CD68+ macrophages predict prognosis [164]. Likewise, in lung-cancer patients, CD103+ lymphocytes have been found to be a positive predicted marker for checkpoint inhibitor therapy responses [165] and found to be associated with a prognostic benefit in cervical-cancer patients [166]. Additionally, the presence of regulatory T cells (Tregs) has been shown to be a negative prognostic biomarker in gynecologic cancer, oral squamous cell carcinoma, and prostate cancer [167,168,169]. In mice bearing melanoma treated with anti-PD-1 immunotherapy, another group has shown that the secretion of INF-y from peripheral lymphocytes can be an accurate biomarker predictive of treatment response [170]. In patients with brain metastases, such as in melanoma, the neutrophil-to-lymphocyte ratio (NLR) has been shown to be a predictive biomarker in response to nivolumab treatment, and a lower ratio was associated with longer overall and progression-free survival [171].

In glioblastoma, one of the most developed areas of biomarker use is that of immunophenotyping, identifying and characterizing components of the immune system within tissue [162]. One group has found, in a mouse model of GBM, that responders to anti-PD-1 treatment combined with radiation increased the CD8+ T-cell-to-Treg ratio [146], and another group demonstrated that blocking CTLA-4 increased CD4+ T-cell proliferative capacity and decreased Treg-mediated immune suppression [172]. More recently, in mice with intracranially implanted gliomas, it has been shown that treatment with temozolomide increases the number of exhausted T cells and ultimately reduces response rates to checkpoint inhibition, leading to the discovery that T=cell exhaustion may be a negative predictive biomarker for ICI [77]. The immune microenvironment can be dominated by T-regulatory cells, and the chemokine CCLS2 by glioma tumors induces this Treg migration [173]. Interestingly another study has shown an increased number of Tregs found in the perivascular space correlates with shorter recurrence time after standard treatments [174]. Finally, in patients treated with intralesional infusion of autologous lymphokine-activated killer cells, activated by incubation with IL-2 prior to infusion into the tumor, improved survival was associated with more frequent CD3+ CD16+ CD56+ lymphokine-activated killer cells, and this was more frequent patients who did not receive corticosteroids in the month prior to treatment [175].

Cytokine production can also be characterized as a biomarker for glioblastoma immunotherapy. In preclinical studies, Wu and colleagues looked at mice implanted with GL261 tumors and treated with combined anti-PD-1 and anti-CXCR4 immunotherapy and found decreased production of pro-inflammatory cytokines, including TNFα and INFγ [176].

Tumor-cell antigens can also be used as biomarkers. In a phase II multicenter clinical trial evaluating an EGFRvIII vaccine, antibodies specific to EGFRvIII were associated with improved overall survival, and loss of tumor EGFRvIII expression was associated with recurrence in 82% of patients [114]. In a phase I trial of patients with newly diagnosed glioblastoma who were treated with a multi-epitope-pulsed DC vaccine, Phuphanich et al. found that the expression of both MAGE1 and AIM-2 was associated with longer PFS and OS, and, similarly, there was a trend toward improved survival with expression of gp100 and HER2 antigens in these patients [177]. Likewise, CD133 expression was noted in areas of tumor recurrence. Nduom et al. examined the prognostic value of PD-L1 and found that high expression levels were associated with worse clinical outcomes [95]. 

Another group identified FCER1G (FcRγ) as a potential biomarker for GBM immunotherapy [178]. This molecule is involved in allergic reactions and encodes the immunoglobulin γ subunit of fragment crystallizable (Fc) region (FcR) [179]. It is involved in immune effector functions, including cytokine release, phagocytosis, oxidative burst, and immune-cell activation [180,181]. FcRγ has been implicated in tumor progression and associated with poorer outcomes when overexpressed. Additionally, in this group’s large cohort of over 1000 patients, it was confirmed that the FCER1G is a novel predictor of patients who respond to immunotherapy effectively [178]. 

Many genetic alterations have been identified in GBM, and some have potential biomarker applications. For example, O6-methylguanine-DNA methyltransferase (MGMT) methylation, while prognostic and potentially predictive of response to temozolomide chemotherapy treatment, has also been associated with improved survival in patients enrolled in GBM vaccine trials, including a phase III trial investigating autologous dendritic cell vaccine therapy [182]. Additionally, mutations in phosphatase and tensin homolog (PTEN) have been associated with non-responders to anti-PD-1 therapy and immune suppression [183]. 

Identifying blood-based biomarkers could be a highly non-invasive way to monitor effectiveness of immunotherapy; however, despite some blood markers being useful in identifying gliomas, such as YKL-40 [184], they lack enough acuity to avoid biopsy or tissue analysis. Moreover, there are currently no true blood biomarkers to identify glioblastoma, although there are blood biomarkers that do correlate with a systemic response to immunotherapy [185,186]. The identification of biomarkers either from cerebrospinal fluid (CSF), peripheral blood, tumor cells, or other tumor-infiltrating cells/tumor microenvironment is an important and currently underdeveloped research area in the field. 

The lack of a clear benefit from GBM immunotherapy, the profound heterogeneity within GBM, and its related microenvironment make finding reliable and consistent biomarkers for immunotherapy response challenging [187]. Currently, while we have associations and predictors of prognosis, there has yet to be discovered a direct 1:1 treatment relationship with a biomarker for the accurate tracking of immunotherapy-treatment effect over time. 

## 6. Conclusions

GBM remains one of the most treatment-resistant cancers to both conventional therapies, such as radiation and chemotherapy, and immunotherapies that have revolutionized treatments of other cancers. The heterogeneity of this disease between patients, the immunosuppressive tumor microenvironment, and the BBB present unique challenges for achieving durable therapeutic responses. The use of immunotherapy to treat GBM has shown significant promise in many preclinical studies; however, this benefit has not yet been realized in larger phase II and phase III clinical trials. Immune checkpoint inhibitors have shown durable responses with the inhibition of PD-1 and CTLA-4 in other cancers; however, when applied to the treatment of GBM, improved outcomes have remained elusive. Understanding the iatrogenic impact of our treatments, most notably dexamethasone immune suppression, will be critical to delivering the most effective immunotherapy. The timing of immunotherapy administration will be just as important as the mechanism of immunotherapeutic action and the combinatorial approach. GBM poses a significant treatment challenge, and we will need a continually deeper understanding of basic immunology, along with autoimmune diseases, such as inflammatory CNS conditions, e.g., multiple sclerosis, and chronic viral infections to better understand immune mechanisms within the brain. As our understanding of the complex microenvironment and new approaches for modulating immune function emerge, rationally designed combinations of immune therapies will hopefully lead to effective therapies that improve patient outcomes by providing long-lasting disease control. 

## Figures and Tables

**Figure 1 biomedicines-10-00427-f001:**
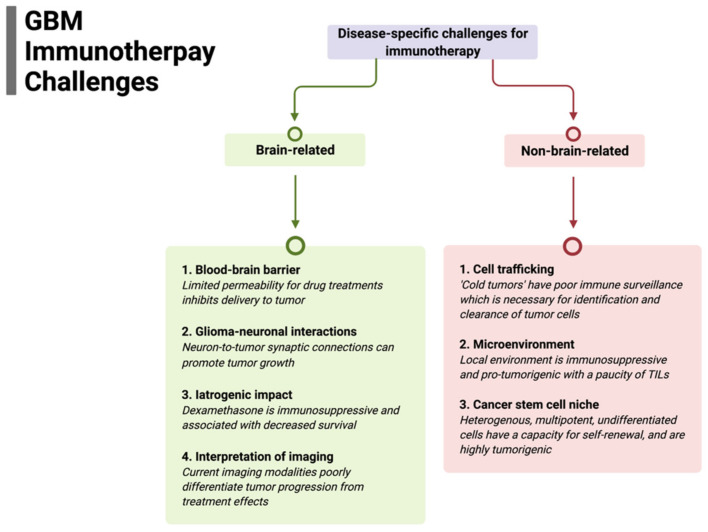
Summary of disease-specific challenges for glioblastoma immunotherapy divided by specific brain-related and non-brain-related challenges that need to be overcome for efficacy.

**Figure 2 biomedicines-10-00427-f002:**
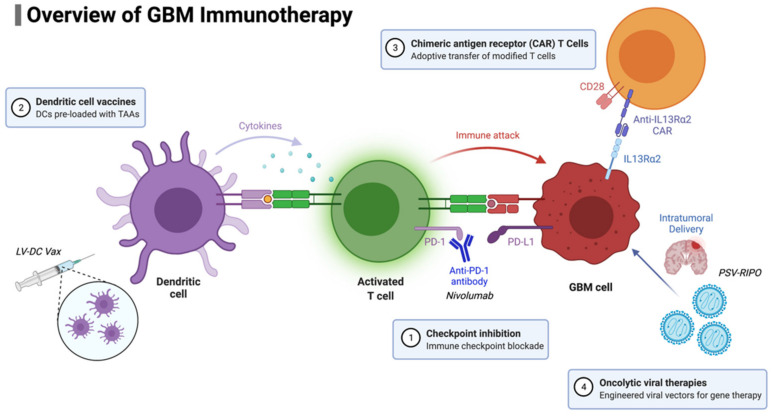
Overview of GBM immunotherapy approaches currently in use and/or development.

**Table 1 biomedicines-10-00427-t001:** Biomarkers for glioblastoma immunotherapy.

Biomarker Type	Subtype	Role/Function	Reference	PMID
Immunophenotyping Biomarkers	CD8+ T cell-to-Treg ratio	Increased in murine responders to combined anti-PD-1 therapy and radiation	Zeng et al.	23462419
	CD4+ T cell proliferation	Enhanced in murine CTLA-4 blockade, along with mitigated Treg-mediated suppression of T cells	Fecci et al.	17404100
	T cell exhaustion	Negative predictor of response to check point inhibition, exhaustion increased with Temozolomide	Karachi et al.	30668768
	CCL2 chemokine	Induces Treg migration into tumor microenvironment	Jordan et al.	17522861
	Perivascular Tregs	Increased number of Tregs in perivascular space associated with shorter time to recurrence	Mut et al.	29163521
	Intralesional CD3+ CD16+ CD56+ lymphokine-activated killer cells	Increased frequency of these cells in patients with primary GBM associated with improved survival, more frequent in patients w/o steroids	Dillman et al.	19816190
Cytokine Biomarkers	TNFa	Decreased in GL261 implanted mice treated with combined anti-PD-1 and anti-CXCR4	Wu et al.	31025274
	INFy	Decreased in GL261 implanted mice treated with combined anti-PD-1 and anti-CXCR4	Wu et al.	31025274
Tumor Cell Antigen Biomarkers	EGFRvIII	Antibodies specific to EGFRvIII associated with improved overall survival, and loss of tumor EGFRvIII expression associated with recurrence	Sampson et al.	20921459
	MAGE1	Increased PFS and OS with tumor expression in phase I trial of multi-epitope-pulsed dendritic cell vaccine	Phuphanich et al.	22847020
	AIM-2	Increased PFS and OS with tumor expression in phase I trial of multi-epitope-pulsed dendritic cell vaccine	Phuphanich et al.	22847020
	gp100	Trend for prolonged survival with tumor expression in phase I trial of multi-epitope-pulsed dendritic cell vaccine	Phuphanich et al.	22847020
	HER2	Trend for prolonged survival with tumor expression in phase I trial of multi-epitope-pulsed dendritic cell vaccine	Phuphanich et al.	22847020
	CD133	Decreased or absent expression in recurrent tumors after treatment with multi-epitope-pulsed dendritic cell vaccine	Phuphanich et al.	22847020
	PD-L1	High expression associated with worse clinical outcomes	Nduom et al.	26323609
	FCER1G	Implicated in tumor progression and associated with poorer outcomes when overexpressed	Xu et al.	33579299
Genetic Biomarkers	MGMT	Methylation associated with improved survival in GBM patients receiving autologous dendritic cell vaccine	Liau et al.	29843811
	PTEN	Mutations have been associated with non-responders to anti-PD-1 therapy and immune suppression	Zhao et al.	30742119
Blood Based Biomarkers	YKL-40	Dramatically elevated in serum of a subset of glioblastoma patients	Tanwar et al.	12154041

## Data Availability

Not applicable.

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
