# Peer review of "Immunotherapeutic Approaches for Glioblastoma Treatment"

_biomedicines, 2022, doi:10.3390/biomedicines10020427_

Round 1

Reviewer 1 Report

The authors have discussed about application of immunotherapies for glioblastoma. Additional suggestions are as follows:

  1. The novelty of the article should be clearly highlighted as number of excellent reviews have already been published on this topic.
  2. The quality of figures is not good and they should be redrawn. New figures should be also added.
  3. Some of the sections of manuscript are too general and add nothing novel to the field such as biomarkers in immunotherapy. More recent references from last two years should be added to improve visibility and quality of current work.
  4. The manuscript should be carefully checked for minor typos and grammatical errors.

Overall, this review appears as a sort of list of known facts, but I'd like that authors express their opinions or critical point of view on the literature. Several points would need to be addressed. Most importantly, I find several passages of the current manuscript very descriptive and thus, it would be important to focus on the conceptual interpretation of findings.  It is important that the review does not end up being an annotated bibliography (i.e. a list of recent findings with no real context or analysis). A review should not only be useful for finding papers, but also push our conceptual understanding forward. Each paragraph should end with a summary and synthesis that links to the main message of your article. This will also help ensure a better flow of the manuscript, connecting the arguments.

Author Response

A point-by-point response to the reviewer’s comments included in uploaded PDF file. Thank you. 

Reviewer 2 Report

The manuscript entitled “Immunotherapeutic approaches for glioblastoma treatment” focuses on a very challenging and interesting topic related with immune-associated therapies in a common adult malignancy of CNS. The authors proceed in a step-wise manner addressing key aspects of this topic. Overall, this review is comprehensive and solid and therefore it expected to be of interest to a range of readers. The following issues could be addressed:

  • The authors could include a Table listing all potential biomarkers for immunotherapy in glioblastoma.
  • Could the authors state the antigens that have been used for the development of vaccines apart from EGFRvIII.

Author Response

(The authors gave the same response as above.)

Round 2

Reviewer 1 Report

The authors have addressed all my concerns.